# Correlation between Serum Zinc Levels and Levodopa in Parkinson’s Disease

**DOI:** 10.3390/nu13114114

**Published:** 2021-11-17

**Authors:** Hirofumi Matsuyama, Keita Matsuura, Hidehiro Ishikawa, Yoshinori Hirata, Natsuko Kato, Atsushi Niwa, Yugo Narita, Hidekazu Tomimoto

**Affiliations:** 1Department of Neurology, Mie University Graduate School of Medicine, 2-174 Edobashi, Tsu 514-8507, Mie, Japan; kmatsuura@clin.medic.mie-u.ac.jp (K.M.); hidehiro-i@clin.medic.mie-u.ac.jp (H.I.); yhirata@clin.medic.mie-u.ac.jp (Y.H.); nkato27@clin.medic.mie-u.ac.jp (N.K.); yug@clin.medic.mie-u.ac.jp (Y.N.); tomimoto@clin.medic.mie-u.ac.jp (H.T.); 2Department of Neurology, National Mie Hospital, 357 Ohsatokubota, Tsu 514-0125, Mie, Japan; niwa.atsushi.yc@mail.hosp.go.jp

**Keywords:** zinc deficiency, zinc-chelating action, dosing frequency of levodopa, psychiatric symptom

## Abstract

Long-term intake of potential zinc-chelating drugs may cause zinc deficiency. We postulated that zinc deficiency in Parkinson’s disease (PD) patients was related to the intake of drugs such as levodopa. We investigated the relationship between zinc levels and levodopa administration period, dosage, and symptoms of zinc deficiency in PD patients. We measured serum zinc levels and analyzed correlations between serum zinc levels, the levodopa oral administration period, dosage, dosing frequency, and zinc deficiency symptoms including taste disorders. Data analyses were performed using Spearman’s rank correlation coefficient. The mean serum zinc level was 60.5 ± 11.6 μg/dL. The mean administration period for levodopa was 8.0 ± 5.5 years, mean administration frequency 3.4 ± 0.9 times/d, and mean administration dose 420.6 ± 237.1 mg/d. Negative correlations between zinc levels and levodopa dosage and dosing frequency were found. Multiple regression analysis showed a significant correlation with the frequency of levodopa (β = −0.360, *p* = 0.007). No significant change in clinical symptoms was observed after zinc administration, but anxiety tended to improve. Our results indicated that frequent levodopa administration strongly influenced serum zinc levels which may have alleviating effects on psychiatric symptoms; therefore, preventing zinc deficiency can be important during PD treatment.

## 1. Introduction

The number of patients with Parkinson’s disease (PD) is constantly increasing, probably due to the aging population; further, there is increasing concern about zinc deficiency in elderly patients. The daily intake of zinc in the elderly is below 50% of the recommended allowance, and the causes include intestinal absorption, inadequate mastication, psychosocial factors, drug interactions, and altered subcellular processes [1]. Drug interaction is one of the causes of zinc deficiency, elderly patients generally have a permanent medication at least once a day, which increases the risk of drug–zinc absorption interactions [2]. In PD patients, long-term intake of potential zinc-chelating drugs, including LCIG (Levodopa–carbidopa intestinal gel) therapy, may cause zinc deficiency [3]. Meanwhile, it has been reported that zinc deficiency may be induced by drugs such as D-penicillamine [4], levodopa, and lithium carbonate [5]. In addition, levodopa has been reported to be a zinc chelator: levodopa can bind to internal zinc, and the resulting compound is excreted in the urine [5]. Over time, this process can lead to zinc deficiency. Generally, the clinical presentation of zinc deficiency includes a taste disorder, anorexia, sexual debility, stomatitis, and dermatitis. More notably, the nervous system and immune systems are particularly affected by zinc deficiency and increased levels of zinc [6]. It has been suggested that zinc homeostasis disturbance has some effect on the development of neurodegenerative diseases such as Alzheimer’s disease, prion disease, and PD [7].

Aside from being a micronutrient involved in skin disorders, wound healing, growth, and immune system activity, among others [8], zinc could improve a person’s sleep quality [9] and can also have psychological effects. Some reports indicated a relationship of zinc levels with psychiatric symptoms such as anxiety [10] and depression [11] among the elderly. In addition, other reports have shown a significant association between the zinc levels in hair and psychiatric symptoms such as depression in patients with PD [12]. Therefore, herein, we measured the serum zinc level of PD patients being treated at our outpatient clinic, and examined the relationship between the duration, dose, and dosing frequency of administration of levodopa preparations, and symptoms of zinc deficiency such as taste disorder, and evaluated the association between zinc status and depression and anxiety in PD.

## 2. Materials and Methods

### 2.1. Patients

In this study, we measured the zinc serum levels of PD patients treated at the Mie University Hospital from February 2018 to August 2018 and examined the relationship between serum zinc levels and the dosage period, dose, and dosing frequency of levodopa, as well as symptoms of zinc deficiency including taste disorder. Patients were diagnosed with PD if they met the requirements for clinically feasible PD, as specified in the Movement Disorder Society Clinical Diagnostic Criteria [13]. Among the patients who agreed to this clinical study, patients with secondary or atypical parkinsonism were excluded. Furthermore, PD patients were assessed for age at onset, disease duration, Hohen and Yahr (H-Y) stage, all parts of MDS-UPDRS score and the Parkinson’s Disease Questionnaire (PDQ) 39 score, including the purpose of assessing the association between serum zinc levels and clinical features such as motor and non-motor symptoms. For patients with advanced PD, the MDS-UPDRS score was evaluated during the patient’s ‘ON’ phase as much as possible. Regarding zinc deficiency, among the clinical symptoms in the diagnostic criteria of zinc deficiency, we confirmed the presence or absence of symptoms for dermatitis, anorexia, sexual debility, stomatitis, and taste disorder [14] through questionnaires at the time of outpatient examination. Since serum zinc levels fluctuate within patients, we fixed the measurement timing in the morning corresponding to the outpatient clinic visit. The reason why, it has been reported that plasma zinc levels fluctuate circadianly, peaking in the morning and troughing in the evening [15]. The serum zinc assessment was conducted with the help of our hospital laboratory experts via venous fasting blood sample. We prespecified the threshold for zinc status using the guidelines for zinc deficiency, with a <80 μg/dL cutoff for low serum zinc levels. Patients with serum zinc <80 μg/dL received oral zinc supplementation for 3 months. A total of 27 patients were reassessed for serum zinc levels after 3 months, after excluding patients who could not continue taking the drug due to side effects, refused to take the drug, and patients who stopped hospital visits. In addition, zinc deficiency symptoms, H-Y stage, MDS-UPDRS score, and PDQ39 score were also re-evaluated after 3 months. The information required to complete this questionnaire was collected by interviewing, reviewing the patients’ medical history, observing, and asking questions to the patients or their family members.

This study was carried out in accordance with the Declaration of Helsinki, and the study protocol approved by the Ethics Committee of the Mie University School of Medicine (H2019-134). Patients provided informed consent in outpatient clinics.

### 2.2. Data Analysis

Statistical analyses were performed using IBM SPSS Statistics 23.0 software (IBM Corporation, New York, NY, USA). Serum zinc levels were not normally distributed in this study population. Therefore, a Mann–Whitney U test was performed for the association between zinc deficiency symptoms and serum zinc levels in patients with PD. The relationship between the levodopa treatment duration, number of years, dose, clinical characteristics (age, PD duration, H-Y stage, MDS-UPDRS score, PDQ39 score) and serum zinc level was evaluated using Spearman’s rank correlation coefficient. In addition, the levodopa dosage period, dose, and dosing frequency were examined by linear regression after adjusting for confounding factors. Regarding the comparison of clinical symptoms before and after zinc administration, nonparametric tests were performed because all parameters (H-Y stage, MDS-UPDRS score, PDQ39 score) had a non-normal distribution. Then, a binomial test was used for the analysis of zinc deficiency (α = 0.05).

## 3. Results

### 3.1. Patient Background Information and Clinical Symptoms

The clinical demographic characteristics of patients are summarized in Table 1. The number of patients with PD was 61(M/F = 33/28, mean age 71.7 ± 8.9 years old), and the average serum zinc level in the population 60.5 μg/mL. The levodopa dosage was 420.6 mg, the dosing frequency 3.4, and the average dosage period 8.0 years. Table 2 shows the relationship between zinc deficiency symptoms and serum zinc levels in patients with PD. When the population was divided based on the presence or absence of zinc deficiency symptoms, the mean serum zinc levels in the group with symptoms were significantly lower than in those without (57.6 ± 10.5 μg/mL vs. 64.6 ± 12.0 μg/mL, respectively; *p* = 0.018).

### 3.2. Correlation between Patient Background and Serum Zinc Levels

We investigated the correlation between patient background and zinc levels. The vertical axis of Figure 1 was the serum zinc level, and the horizontal axis was the levodopa dosage, frequency, and dosage period. Spearman’s rank correlation showed a significant inverse correlation of serum zinc levels with levodopa dosage (rs = −0.290, *p* = 0.023) and frequency (rs = −0.454, *p* = 0.000). However, there was an inverse correlation between serum zinc levels and age (rs = −0.276, *p* = 0.031). No clear correlation was found for the other factors (age at onset, PD duration, H-Y stage, MDS-UPDRS, PDQ39 scores). As we considered the possibility of a significant relationship between serum zinc levels and age in PD patients, we performed an age-adjusted multiple regression analysis, including zinc level as the objective variable, and age, PD duration, and H-Y stage as explanatory variables. Although no significant interaction effect was observed from levodopa dosage (β = −0.211, *p* = 0.111; Table 3), a significant correlation was found with levodopa frequency (β = −0.360, *p* = 0.007; Table 3). 

### 3.3. Symptoms before and after Oral Zinc Administration

We also investigated patients’ symptoms before and after oral zinc administration. Out of the 36 patients with a serum zinc level < 80 μg/dL and showing symptoms, 27 received oral zinc preparations (Figure 2). Nine patients were unable to continue taking the drug, because of side effects, a refusal to take the drug, stopping to visit the outpatient clinic, etc. Of the 27 patients receiving zinc supplementation, 15 had improved on their symptoms which included taste disorder and dermatitis, but no significant improvement in MDS-UPDRS, PDQ39 score, and H-Y stage was shown after supplementation in any of the 27 patients. A binomial test showed that taste disorder and dermatitis significantly improved after zinc supplementation (respectively; *p* = 0.011, *p* = 0.016; Table 4). Although not significant, MDS-UPDRS part I showed a tendency towards improvement in some patients. A single item analysis on MDS-UPDRS part I showed a tendency toward improvement of the anxious mood (*p* = 0.052; Table 5).

## 4. Discussion

In our study, we found that low serum zinc levels in PD patients were probably influenced by levodopa administration. Serum zinc levels were negatively correlated with levodopa dosage and frequency. In particular, multiple regression analysis revealed that the frequency of levodopa use had a strong effect on serum zinc levels. Further, zinc supplementation could improve taste disorder and dermatitis, but there was no significant effect on clinical symptoms as indicated by MDS-UPDRS, PQQ39 scores and H-Y stage. In some cases, zinc supplementation also improved anxiety and was suggested to have some effect on psychiatric symptoms.

The causes of zinc deficiency include insufficient zinc intake, inadequate absorption, increased demand, and increased excretion, which is attributed to continuous administration of chelating drugs; levodopa being a known drug having this effect. Drugs with ≥2 hydroxide ions, such as levodopa, may chelate zinc and stabilize it by forming a five- or six-membered ring [16]; these chelated compounds are in turn excreted in the urine, which may result in zinc deficiency. Previous studies of serum zinc concentration in drug chelates have already been investigated in rats. Tetracycline was also known to chelate with zinc such as levodopa, and it has been reported that an increase in serum zinc level was significantly suppressed as the dose up of tetracycline orally administered at the same time as zinc, indicating the effect of drug interaction [5,17]. In this study, Spearman’s analysis showed an inverse correlation between serum zinc levels and the levodopa dosage, suggesting drug chelation. Since levodopa and tetracycline are different drugs, it is not possible to compare our study with Penttilä et al.’s study [17], but a similar mechanism can be considered in that chelation lowers serum zinc levels. When age and H-Y stage were included in the objective variables in multiple regression analysis, the interaction effect between levodopa dosage and serum zinc level was not significant, but the interaction with age remained significant. In Japanese adults, a highly significant inverse correlation was observed between age and zinc concentration in hair [18], which led us to consider an age-related decrease in zinc concentration in our study. However, even if age was included in the explanatory variables in multiple regression analysis, the significant negative correlation in levodopa dosing frequency was maintained. This suggested that serum zinc levels in PD patients are strongly influenced by the frequent levodopa administration. The introduction of LCIG therapy significantly improved specific nonmotor symptoms in advanced PD patients, enhanced quality of life, and facilitated daily living activities as shown by Krüger et al. [19]. On the other hand, we have previously reported that advanced zinc deficiency was associated with the first introduction of LCIG therapy [3]. Already then, we postulated that the ability of the intestinal mucosa to absorb zinc may be disrupted by continuous levodopa administration. In addition, our previous result is in accordance with the findings of this study where frequent doses of levodopa had a strong lowering effect on serum zinc.

Decreased zinc in plasma or serum of PD patients has been previously reported in a small population [20,21]. Further, Zhao et al. [22] found that copper and zinc concentrations were reduced in PD patients compared with controls, showing that lower plasma zinc levels were associated with an increased risk of PD [22], and Kim et al. [23] investigated the relationship between the blood levels of zinc, iron, and copper in 325 PD patients and clinical data such as movement disorders, cognitive function, severity, and levodopa administration [23]. In this report, there was no significant association between serum zinc levels and clinical symptoms such as dyskinesia and motor fluctuation, similar to our research results. To our knowledge, this is the first report evaluating levodopa as the reason for low zinc levels in PD patients.

Recent studies have further proposed a mechanism between zinc and psychiatric disorders [24,25,26]. Presumably due to zinc being involved in the regulation of brain receptors such as serotonin [27] and glutamate receptors [28], although another study found that zinc may be associated with psychiatric disorders due to its anti-inflammatory and antioxidant effects [29]. Our study could not show more than an association between zinc deficiency and psychiatric symptoms and therefore, we cannot assess causality. However, as our results suggest an improvement of anxiety with zinc supplementation; this should be further assessed in the future. Our study had some limitations. First, the sample size was relatively small and the statistical power was limited; therefore, the interaction between zinc and sex was not investigated. Second, we did not investigate zinc concentration in the urine and could not explain the mechanism of increased zinc excretion. Third, we did not compare this population with healthy subjects, nor did we compare with PD patients who do not take levodopa. Fourth, there is no blinding when evaluating data from this population. This is considered a weakness in the data in this paper. Fifth, this time we have not examined the diet and exercise habits of the population, so we have not been able to confirm whether there is a correlation with zinc level. However, we consider that these data are an essential first step to consider the clinical implications of zinc deficiency in PD patients. A further large-scale study including the above investigations is warranted.

## 5. Conclusions

This is the first study to show the significant effect of the frequency of levodopa administration in PD patients on serum zinc levels. Further, zinc deficiency tended to have psychological effects.

## Figures and Tables

**Figure 1 nutrients-13-04114-f001:**
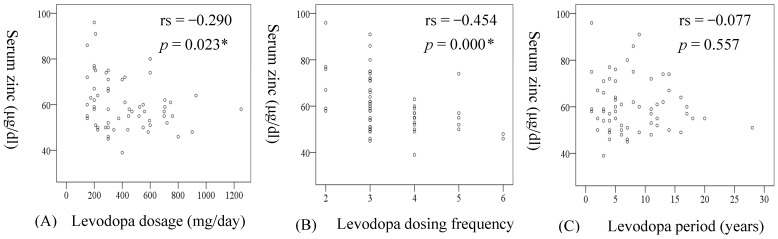
Correlation between patient background and serum zinc level. A significant negative relationship with serum zinc levels was found in levodopa dosage (**A**: rs = −0.290, *p* = 0.023) and dosing frequency (**B**: rs = 0.454, *p* = 0.000), but none was found in the levodopa period (**C**: rs = −0.077, *p* = 0.557). * *p* < 0.05.

**Figure 2 nutrients-13-04114-f002:**
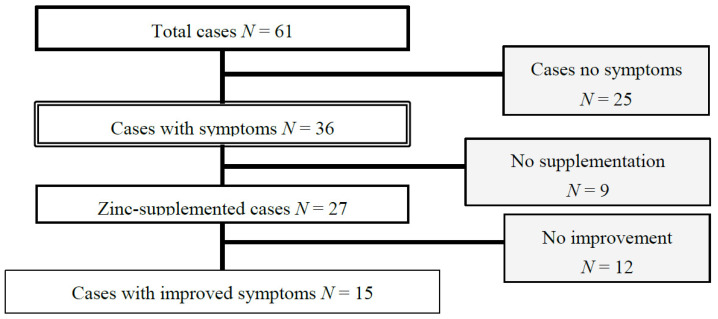
Symptoms before and after oral administration of zinc. In total, 27 cases with serum zinc less than 80 and symptom were supplemented with zinc, there were 15 cases in which zinc deficiency were improved.

**Table 1 nutrients-13-04114-t001:** PD patient demographicsand clinical data.

	PD Patients
Characteristic	(*N* = 61)
Age, years	71.7 ± 8.9
Male (%)	33 (54.1)
Age at onset (years)	61.2 ± 11.2
PD duration (years)	10.5 ± 6.4
Modified H-Y stage	2.6 ± 0.9
MDS-UPDRS Part I	9.1 ± 6.1
Part II	10.7 ± 7.4
Part III	20.3 ± 12.0
Part IV	1.8 ± 2.4
PDQ 39	115.1 ± 25.6
Serum zinc (μg/mL)	60.5 ± 11.6
L-dopa dosage (mg)	420.6 ± 237.1
L-dopa dosing frequency (times)	3.4 ± 0.9
L-dopa period (years)	8.0 ± 5.5
Zinc deficiency (%)	
taste disorder, *N*	14 (22.3)
dermatitis, *N*	16 (26.2)
sexual debility, *N*	3 (4.9)
anorexia, *N*	8 (13.1)
stomatitis, *N*	7 (11.5)
Data are presented as mean ± SD, or number (percentage).	

PD, Parkinson’s disease; PDQ 39, Parkinson’s Disease Questionnaire; H-Y stage, Hoehn and Yahr stage; MDS-UPDRS, MDS-Unified Parkinson’s Disease Rating Scale.

**Table 2 nutrients-13-04114-t002:** Association of Zinc Deficiency Symptoms and Serum Zinc Levels in PD patients.

Symptom of Zinc Deficiency	with Symptom	No Symptom	*p* Value
	*N* (Zinc Levels, ug/mL)	*N* (Zinc Levels, ug/mL)	
taste disorder	14 (57.3 ± 9.9)	47 (61.4 ± 12.0)	0.425
anorexia	8 (55.3 ± 8.9)	53 (61.2 ± 11.9)	0.111
dermatitis	16 (54.0 ± 6.3)	45 (62.8 ± 12.3)	0.01 *
sexual debility	3 (77.3 ± 16.7)	58 (59.6 ± 10.8)	0.036 *
stomatitis	7 (56.6 ± 8.6)	54 (61.0 ± 11.9)	0.417
Total	36 (57.6 ± 10.5)	25 (64.6 ± 12.0)	0.018 *

* *p* < 0.05.

**Table 3 nutrients-13-04114-t003:** Multiple regression analysis with zinc levels as objective variable in PD patients. Age, PD duration, and H-Y stage were added to the explanatory variables for analysis for each of levodopa dose, frequency, and levodopa duration.

Explanatory Variable	*β*
levodopa dosage	−0.21
age	−0.30 *
PD duration	−0.09
H–Y stage	0.01
*R* ^2^	0.16
Explanatory variable	*β*
levodopa frequency	−0.36 **
age	−0.26
PD duration	−0.08
H–Y stage	0.01
*R* ^2^	0.25
Explanatory variable	*β*
levodopa duration	−0.2
age	−0.35 *
PD duration	−0.02
H–Y stage	0.01
*R* ^2^	0.13

PD, Parkinson’s disease. * *p* < 0.05, ** *p* < 0.01. *β,* Standardized partial regression coefficient.

**Table 4 nutrients-13-04114-t004:** Symptoms before and after oral zinc administration.

*N* = 27	Before Supplement	After Supplement	*p* Value
serum zinc (μg/mL)	55.9 ± 7.8	79.5 ± 12.1	
taste disorder, *N*	10 (37.0%)	4 (14.8%)	0.011 *
dermatitis, *N*	12 (44.4%)	6 (22.2%)	0.016 *
sexual debility, *N*	2 (7.4%)	2 (7.4%)	1.000
anorexia, *N*	6 (22.2%)	3 (11.1%)	0.125
stomatitis, *N*	7 (25.9%)	6 (22.2%)	0.423
MDS-UPDRS Part I	10.3 ± 6.1	8.7 ± 4.9	0.335
Part II	9.7 ± 6.6	10.3 ± 8.9	0.796
Part III	20.5 ± 10.7	20.5 ± 11.4	1.000
Part IV	1.7 ± 2.4	1.7 ± 2.3	0.937
PDQ39	115.0 ± 22.8	117.6 ± 24.2	0.715

* *p* < 0.05.

**Table 5 nutrients-13-04114-t005:** Symptoms before and after oral zinc administration with MDS-UPDRS score Part I.

Questions in Part I	Before Supplement	After Supplement	*p* Value
1.1	0.52 ± 1.01	0.54 ± 0.95	0.895
1.2	0.85 ± 1.23	0.69 ± 1.19	0.519
1.3	0.93 ± 1.04	0.58 ± 0.86	0.198
1.4	1.00 ± 0.96	0.54 ± 0.76	0.052
1.5	0.52 ± 0.94	0.50 ± 0.81	0.821
1.6	0.26 ± 0.45	0.23 ± 0.43	0.811
1.7	1.04 ± 0.90	1.00 ± 1.10	0.643
1.8	1.04 ± 0.90	1.15 ± 0.67	0.543
1.9	0.81 ± 0.88	0.65 ± 0.69	0.602
1.1	1.26 ± 1.20	0.85 ± 0.73	0.314
1.11	0.93 ± 0.73	0.85 ± 0.83	0.662
1.12	0.41 ± 0.64	0.50 ± 0.81	0.831
1.13	0.70 ± 0.72	0.58 ± 0.64	0.55

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
