# Peer review of "Correlation between Serum Zinc Levels and Levodopa in Parkinson’s Disease"

_nutrients, 2021, doi:10.3390/nu13114114_

Round 1

Reviewer 1 Report

I think overall the study was very interesting and conclusions were sound. I think some additional details are needed but these should be fairly easy to add. I think there were a couple things that were confusing but may be more of an issue of language than content. In the introduction I think better to state Duodopa than device therapy as the later suggests possibly DBS. In the introduction I think there is probably a wider array of literature but just positive studies quoted. Say our hospital but then talk about out patient - not sure if hospital or outpatient clinic. Line 62 I am not sure what "patient's timing as much as possible" means. Should include a consort diagram for those not enrolled. 104 that should be those. Lack of blinding should be included as a weakness. There was a mentioning of timing of zinc measurement being significant this could be detailed more. 

Reviewer 2 Report

This is an interesting and timely study that could have important implications for Parkinson disease treatment. 

I will include some comments: 

Please enlarge introduction, including the effect of zinc in immune system and sleep ( see following papers), and the previous data on zinc level in pathogenesis of neurodegenerative diseases, such as Parkinson .

Zinc and the immune system.

Rink L, Gabriel P.

Proc Nutr Soc. 2000 Nov;59(4):541-52. doi: 10.1017/s0029665100000781.

Zinc: dietary intake and impact of supplementation on immune function in elderly.

Mocchegiani E, R et al.

Zinc Acts as a Sleep Modulator.

Cherasse Y, Urade Y.

Int J Mol Sci. 2017 Nov 5;18(11):2334. doi: 10.3390/ijms18112334.

Results:

If you have data on sleep/restlesslegs syndrome in your data sets, would be interesting to correlate it among symptoms. 

As a further note, it would have been of interest to investigate diet through a diary and style of life (exercise) to see if there is a correlation with zinc level, also. I can imagine that this is also an issue, except than age

The authors correctly stated that the data were not controlled with PD patients who do not take levodopa. I suggest in the next papers, to confront also zinc level in dopamine-agonist treated patients (three groups: PD with no levodopa; PD with levodopa; PD with dopamine-agonist). 

Please, improve English especially fluency in discussion,
